# Compound mortality impacts from extreme temperatures and the COVID-19 pandemic

Y. T. Eunice Lo ●[1,2] ✉, Dann M. Mitchell ●[1,3] & Antonio Gasparrini ●[4]

Extreme weather and coronavirus-type pandemics are both leading global health concerns. Until now, no study has quantified the compound health consequences of the co-occurrence of them. We estimate the mortality attributable to extreme heat and cold events, which dominate the UK health burden from weather hazards, in England and Wales in the period 2020-2022, during which the COVID-19 pandemic peaked in terms of mortality. We show that temperature-related mortality exceeded COVID-19 mortality by 8% in South West England. Combined, extreme temperatures and COVID-19 led to 19 (95% confidence interval: 16–22 in North West England) to 24 (95% confidence interval: 20–29 in Wales) excess deaths per 100,000 population during heatwaves, and 80 (95% confidence interval: 75–86 in Yorkshire and the Humber) to 127 (95% confidence interval: 123–132 in East of England) excess deaths per 100,000 population during cold snaps. These numbers are at least ~2 times higher than the previous decade. Society must increase preparedness for compound health crises such as extreme weather coinciding with pandemics.

Climate change is deeply embedded in the long-term global health concern[1], with much of its direct health burden coming from extreme weather events. However, it is a crisis that is nearly always considered in isolation from other crises, even though there can be compound socio-economic impacts when major crises occur in parallel. While the climate-conflict nexus is relatively well studied[2,3], other parallel crises such as climate change and obesity, climate change and artificial intelligence, and, as studied here, extreme weather and pandemics, are far less investigated.

Extreme weather events are "shattering" observational records, disrupting infrastructure, and, in the worst cases, claiming lives. In the period that the most recent global pandemic – COVID-19 – was officially declared a public health emergency of international concern by the World Health Organisation (WHO; between January 2020 and March 2023), the UK recorded 40.3 °C unprecedented extreme heat[4] and a record number of around 3000 heatwave excess deaths[5], western North America experienced one of the most extreme heatwaves ever recorded globally[6] that led to hundreds of deaths[7], Pakistan had floods that killed nearly 2000 people and displaced 7 million more[8], and the Horn of Africa experienced persistent droughts that led to

over 3 million people requiring humanitarian assistance[9]. These places have different levels of death registration completeness[10] and cause of death accuracy[11], so these numbers cannot be compared like-for-like. However, the life-threatening weather events, attributable in part to human-induced climate change[12–14], prompted scientists to warn about compound societal risks arising from the intersection of climate hazards and pandemics like COVID-19[15].

One way the intersection of extreme weather events and pandemics can result in major societal risks is through overwhelming health systems. If health services are already operating at capacity because of one crisis, the additional health burden from another crisis can break the system entirely, endangering the lives of many people. In the two years of 2020-2021, almost 15 million excess deaths were estimated to be associated with COVID-19 directly or indirectly globally[16]. From the emergence of COVID-19 in December 2019 to the writing of this paper in late 2023, about 230,000 people in the UK, or 0.3% of its total population, have died from COVID-19 according to death certificates[17]. These numbers would likely have been much higher had there not been preventative measures put in place, such as physical distancing or vaccination[18–20]. The rapid development,

[1]Cabot Institute for the Environment, University of Bristol, Bristol, UK. [2]Elizabeth Blackwell Institute for Health Research, University of Bristol, Bristol, UK. [3]School of Geographical Sciences, University of Bristol, Bristol, UK. [4]Environment & Health Modelling (EHM) Lab, Department of Public Health Environments and Society, London School of Hygiene & Tropical Medicine, London, UK. ✉e-mail: eunice.lo@bristol.ac.uk

severity, and adverse impacts of the pandemic on public health and the functioning of society meant that it dominated global news and everyday life for an extended period of time[21].

However, the exact respective and compound health impacts of extreme weather and COVID-19 remain unknown. Understanding these impacts from observations of recent events is extremely important to inform climate adaptation, long-term pandemic preparedness, and public health solutions that simultaneously address both[15].

Here, we use England and Wales as a well-observed and well-documented case study to illustrate the respective and combined mortality impacts of weather hazards and COVID-19, in the period between 30 January 2020 and 31 December 2022. The first COVID-19 death recorded on a death certificate in England and Wales was on 30 January 2020 (in South East England), so this date is chosen as the starting date for this study. We chose England and Wales because we already have location-specific epidemiological models linking temperature variability to daily mortality set up for regions within them[22], and that daily COVID-19 mortality statistics for the same sub-national regions are publicly available from the UK Government dashboard[17].

We consider excess mortality associated with non-optimal high and low temperatures because it dominates the short-term health impact of other weather hazards[23], including floods, storms, and droughts in the UK. Temperature-related mortality is also relatively straightforward to quantify through time series epidemiological modelling. Specifically, we focus on the hottest (heatwaves) and coldest (cold snaps) days during the study period, to understand the compound impacts when regional extreme temperatures coincide with the global pandemic.

## Results

Figure 1a shows that heat-related mortality (red lines, with red shading indicating its 95% confidence interval) in England and Wales primarily occurred between July and September during the study period of 30 January 2020 to 31 December 2022. A total of 8481 excess deaths (95% confidence interval: 6387–10,493) were attributable to high temperatures, with daily heat-related mortality peaking at 580 deaths (95% confidence interval: 484–670) on 19 July 2022, when England recorded 40.3 °C unprecedented extreme heat. Figure 1b zooms in on the time evolution of mortality during this heatwave for greater legibility. Although the UK Met Office and UK Health Security Agency (UKHSA) have issued Level 2 (yellow) and Level 3 (amber) Heat Health Alerts for all regions except North East England since 11 July 2021 to warn the public about this heatwave, which they subsequently raised to the highest Level 4 (red) alert for all regions on 18 July 2021 to trigger a national emergency response[5], peak heat-related mortality during this heatwave exceeded the maximum daily level from the previous ten years (i.e., 2010-2019; red error bar in Fig. 1a), which was 376 deaths (95% confidence interval: 304–445).

In months other than July, August and September, cold-related mortality (blue lines, with blue shading indicating its 95% confidence interval) dominated over heat-related mortality. Over the study period, a total of 128,533 excess deaths (95% confidence interval: 107,430–153,642) were attributable to low temperatures, indicating a fifteen-fold larger cold-than-heat mortality burden. These results are consistent with the literature, which found that most days of the year are considered moderately cold in England and Wales, resulting in a large number of cold-related deaths[24]. Daily cold-related mortality peaked at 531 excess deaths (95% confidence interval: 493–574) on 15 December 2022, but this falls within the range from the previous ten years (maximum: 691 deaths, 95% confidence interval: 643–743).

These temperature-related deaths are theoretically independent of COVID-19 deaths because they are calculated from distributed lag nonlinear models (DLNMs) that describe the relationships between daily mean temperature and daily all-cause mortality after COVID-19

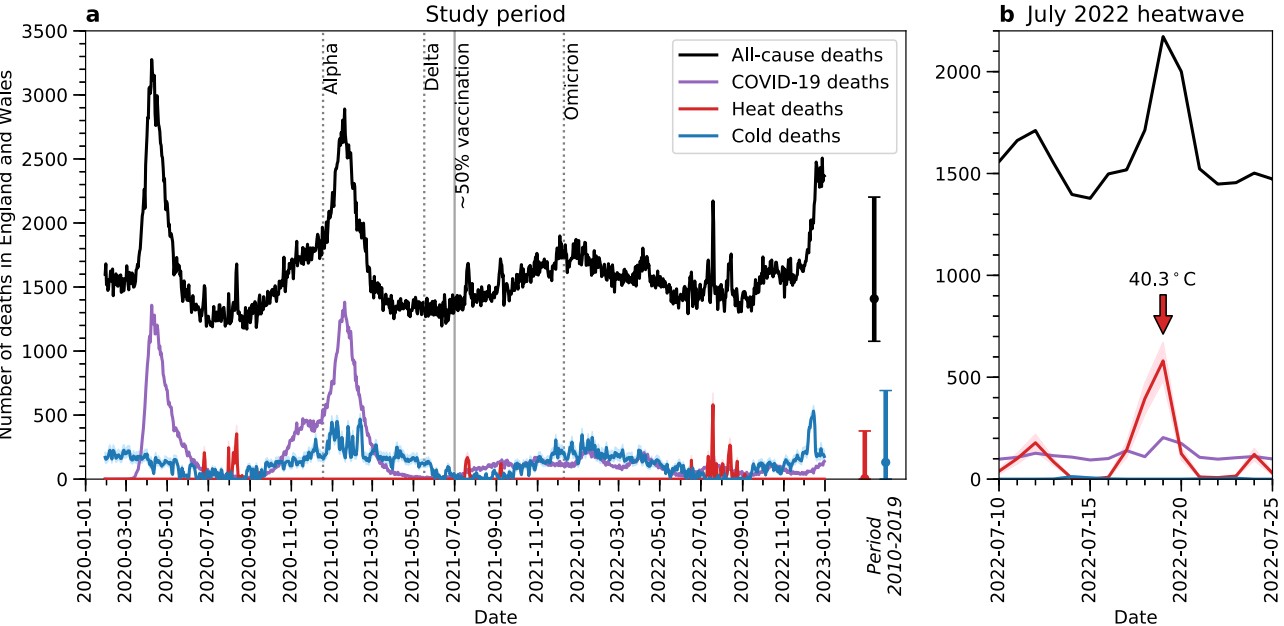

**Fig. 1 | Time series of daily deaths attributable to non-optimal temperatures, COVID-19, and all causes in England and Wales for the same period.** In both panels, red lines indicate the best estimate of heat-related deaths, with red shading indicating its 95% confidence interval. Blue lines indicate the best estimate of cold-related deaths, with blue shading indicating its 95% confidence interval. All temperature-related deaths here represent the sum of regional deaths estimated from individual temperature-mortality associations for ten regions in England and Wales (see "Methods"). Purple lines indicate COVID-19 deaths as shown on death certificates. Black lines indicate the total number of deaths from all causes. In panel (**a**) dotted vertical lines indicate the time of emergence of dominant COVID-19 variants in the UK during the study period of 30 January 2020 to 31 December 2022. The solid vertical line indicates the time around which 50% of the population aged 12 or above have received two doses of COVID-19 vaccines in regions within England and Wales. The error bars in panel **a** indicate the range of heat-related, cold-related, and all-cause deaths in the period 1 January 2010 to 31 December 2019, with the centre points indicating the mean values. In panel (**b**), which zooms in on the July 2022 UK heatwave, the red arrow indicates the date on which 40.3 °C was recorded.

mortality has been removed (see Methods). For comparison, Fig. 1 also shows the time evolution of daily COVID-19 mortality according to death certificates (purple lines). Distinct surges in COVID-19 deaths were seen soon after the first emergence of COVID-19 in early 2020 and the domination of the Alpha variant in December 2020 (dotted vertical line in Fig. 1a)[25], with the highest daily mortality level being 1382 deaths on 19 January 2021. The emergence of the Delta and Omicron variants was not followed by as large a surge in deaths, likely because COVID-19 vaccination had become more common by then (see Figure S1), with about half of the population above the age of 12 having had two doses of COVID-19 vaccines by 1 July 2021 (solid vertical line in Fig. 1a), in all regions in England and Wales except London (see Table S1). Over the whole study period, 194,480 COVID-19 deaths were reported on death certificates in England and Wales.

Figure 1a shows that from June to October 2020, March to August 2021, and from September 2021 to the end of 2022, temperature-related deaths (the sum of heat- and cold-related deaths) exceeded COVID-19 deaths. These exceedances were driven by heat-related mortality spikes when COVID-19 mortality was relatively low, e.g., during the July 2022 heatwave (Fig. 1b), as well as cold-related mortality dominating in the colder months after COVID-19 vaccination was introduced. To further examine the respective mortality impacts of non-optimal temperatures and COVID-19, Fig. 2 shows the ratios of cumulative deaths from these two causes for each region in England and Wales, across the whole study period (panel a), and during heatwaves and cold snaps therein (panels b and c).

Considering the whole study period of 30 January 2020 to 31 December 2022, cumulative temperature-related deaths exceeded cumulative COVID-19 deaths by 8% in South West England. While this exceedance did not occur in the other regions, temperature-related deaths amounted to 58% (East Midlands) to 75% (London) of COVID-19 deaths by the end of 2022. These results demonstrate the importance of increasing public health messaging about heat and cold, which tends to be far less prevalent than the messaging about COVID-19. Reducing temperature-related mortality would free up resources and capacity for health services to respond to major pandemics when they occur.

Since extreme weather events are where we would expect the health effects to be largest, focusing on them provides important information on their interplay with other parallel health crises, including their compound health effects. Figure 2b shows that during the ten heatwave episodes (spanning a total of 70 days; see Table S2) in the study period, identified through UKHSA's Heat Mortality

Monitoring Reports[5,26,27], temperature-related deaths outnumbered COVID-19 deaths in 9 of the 10 regions (except in North West England). This exceedance is particularly apparent in the southern regions where heat stress is more pronounced[28]. The ratios of temperature-related deaths to COVID-19 deaths in the southern regions range from 1.7 in East of England and South East England to 2.7 in London. The ratios for the rest of the regions lie between 1.1 and 1.3, except for North West England which has a ratio of 0.8. These results highlight that even during the COVID-19 pandemic, heatwaves posed a serious threat to public health, which is often downplayed[29] or misrepresented as something enjoyable by the media in the UK[30].

Figure 2c shows the corresponding results during eight cold snaps in the study period, which are defined here as days on which a Level 3 Cold Health Alert was issued by UKHSA for any region in England (also spanning 70 days; see Table S2). A Level 3 (amber) Cold Health Alert represents a situation in which impacts are likely to be felt across the health and social care sectors, and potentially the whole population[31]. During these cold snaps, temperature-related deaths were lower than COVID-19 deaths in all regions, with the ratios ranging from 0.4 in East of England to 0.8 in South West England. These results are likely to be driven by the large surges in COVID-19 mortality following the first emergence of the coronavirus and the domination of the Alpha variant, both of which occurred in winter (Fig. 1a and S1). In this sense, our results should not be interpreted as low temperatures being less important than COVID-19 to health in winter, as we have already shown that cold-related mortality occurs throughout the year and dominated over COVID-19 in the second half of the study period (Fig. 1a). Future outbreaks of COVID-19 or novel viruses could have a different seasonal pattern from the COVID-19 pandemic studied here. Therefore, they could have different health impacts relative to extreme cold in winter.

The co-occurrence of non-optimal temperatures and COVID-19 meant that all-cause mortality in England and Wales was, on average, higher in the study period than in the previous ten years (black line and bar in Fig. 1a). During extreme events, the health system needed to deal with an unprecedented compound health impact from both extreme weather and COVID-19. Figure 3a shows the total number of deaths arising from high temperatures and COVID-19 during the 70 heatwaves days in the study period. Regional compound (heat-related and COVID-19) mortality ranged from 19 deaths per 100,000 people (95% confidence interval: 16–22) in North West England, to 24 deaths per 100,000 people (95% confidence interval: 20–29) in Wales.

These compound mortality levels are put into context by comparing Fig. 3a with Fig. 3c, which shows the reference levels of heat-

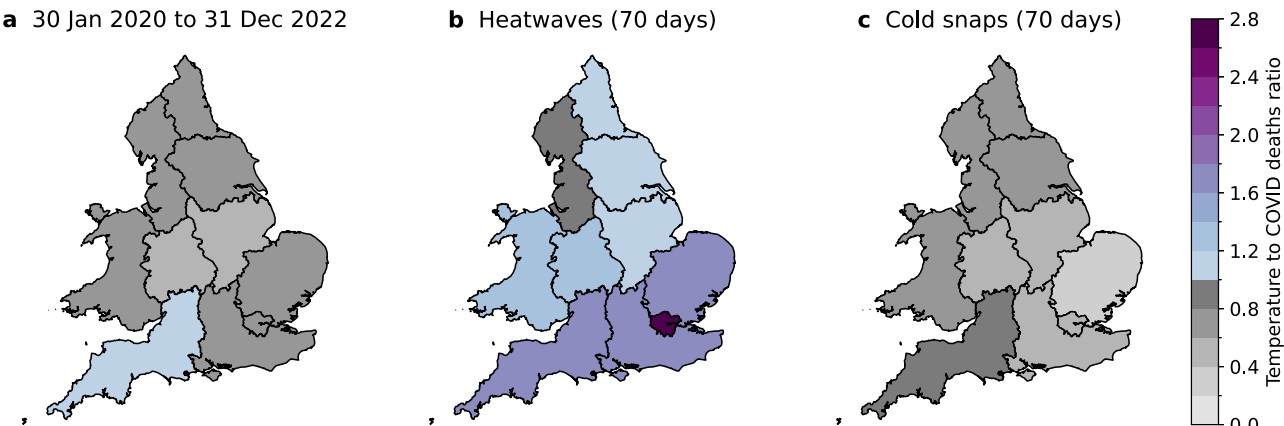

**Fig. 2 | Maps of the ratio of temperature-related deaths to COVID-19 deaths during extreme temperature events.** Panel **a** shows regional ratios for the whole study period, i.e., 30 January 2020 to 31 December 2022. Panel **b** shows regional ratios on 70 heatwave days (in a total of 10 heatwaves) during the study period.

Heatwaves are defined following the UKHSA definition. Panel **c** shows regional ratios on 70 cold snap days (in a total of 8 cold snaps) during the study period. Cold snaps are defined as days on which a Level 3 Cold Health Alert was issued for any region in England.

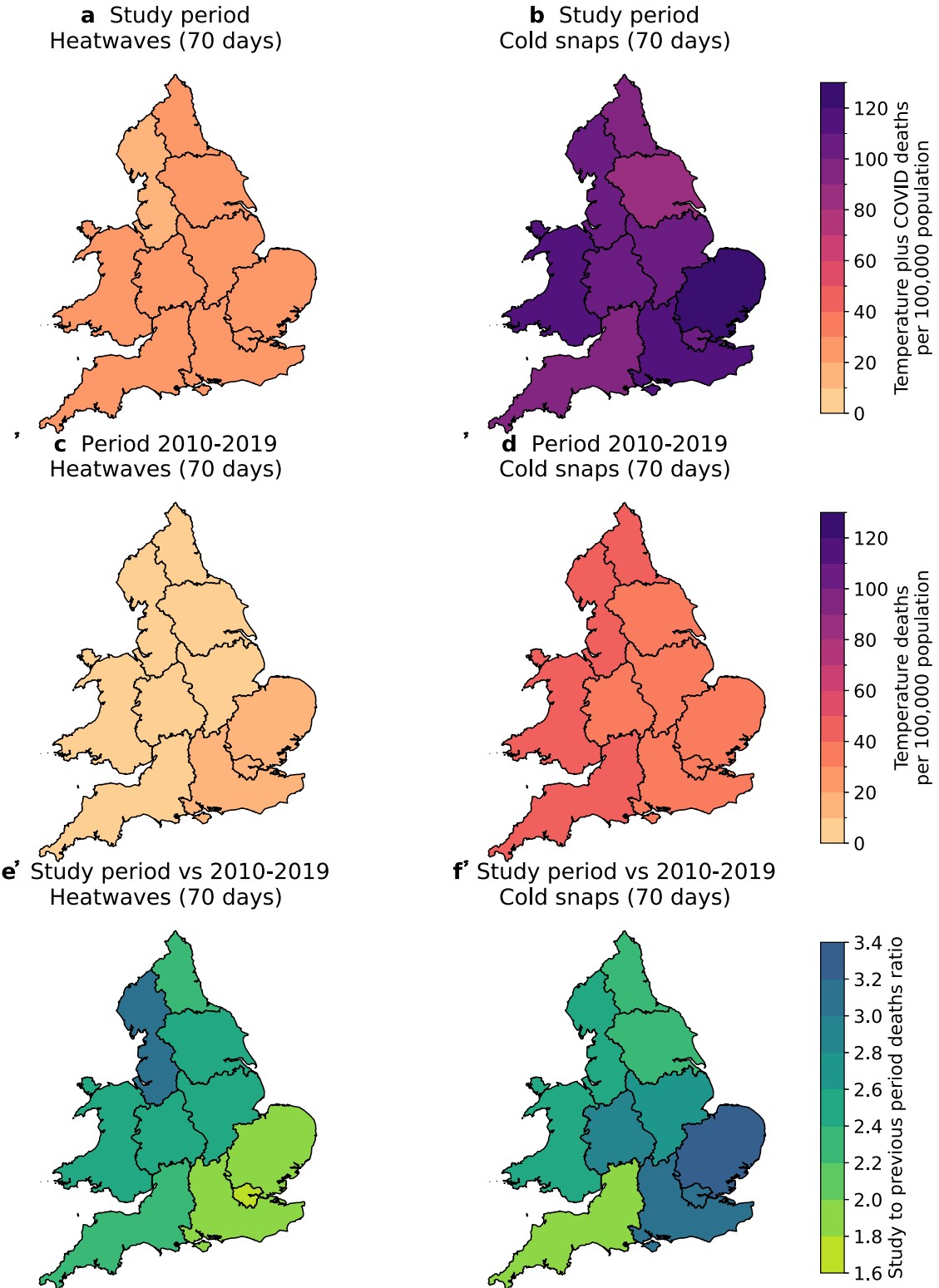

**Fig. 3 | Maps of the total number of deaths per 100,000 population from non-optimal temperatures and COVID-19 during extreme temperature events, and the comparison of these number before and during the study period.** Panels (**a**) and (**b**) show regional sums of temperature-related and COVID-19 deaths on 70 heatwave days (in a total of 10 heatwaves; panel (**a**)) and 70 cold snap days (in a total of 8 cold snaps; panel (**b**)), during the study period of 30 January 2020 to 31 December 2022. Panels (**c**) and (**d**) show regional numbers of temperature-related deaths on the same number of heatwaves (panel (**c**)) and cold snap days (panel (**d**)) but from the period 2010–2019. The numbers in panels **c** and **d** are estimated from the average number of temperature-related deaths per heatwave or cold snap day in the period 2010–2019, multiplied by 70 days. Panels (**e**) and (**f**) show regional ratios of deaths during the study period to the 2010–2019 period for heatwaves (panel (**e**)) and cold snaps (panel (**f**)).

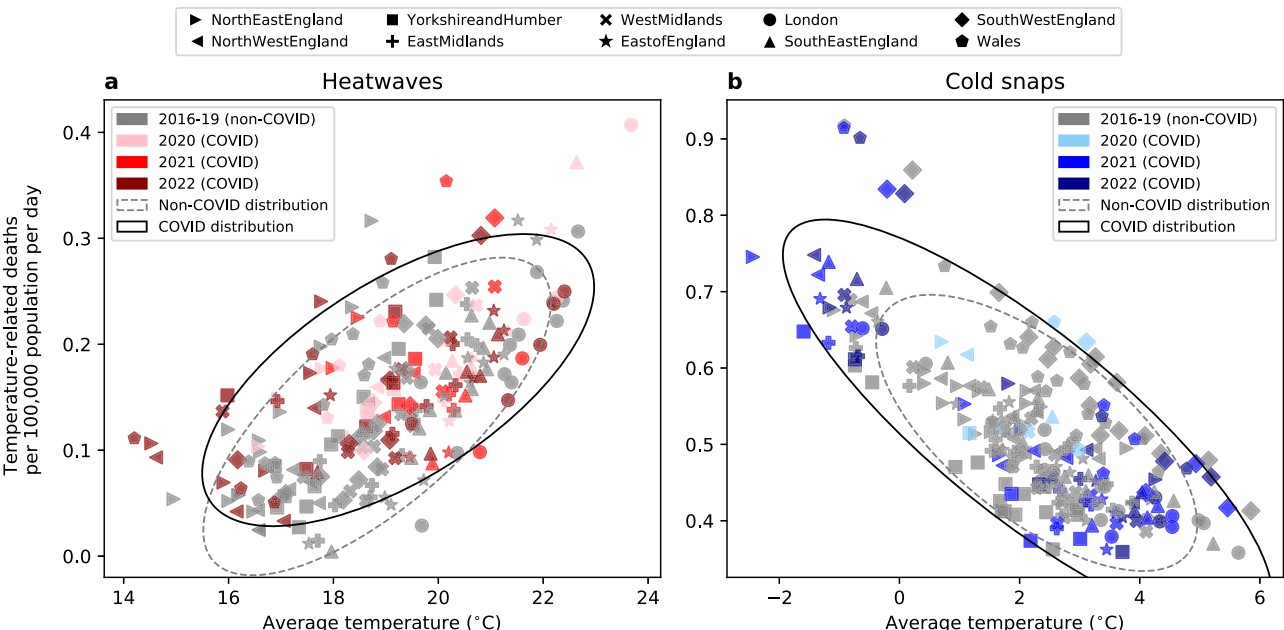

**Fig. 4 | Average temperature-related deaths per 100,000 population per day versus the average temperature of individual heatwaves and cold snaps in 2016-2022.** The markers indicate the regions. In panel (**a**), grey markers indicate heatwaves in 2016–2019, whereas coloured markers indicate heatwaves in COVID-19 affected years: 2020 (pink), 2021 (red) and 2022 (dark red). In panel (**b**), grey markers indicate cold snaps in 2016–2019, whereas coloured markers indicate cold snaps in COVID-19-affected years: 2020 (light blue), 2021 (blue) and 2022 (dark blue). In both panels, the grey dashed ellipses indicate the two standard deviation confidence of the covariance of mortality and temperature of all heatwaves or cold snaps in the non-COVID-19 period of 2016-2019. The black solid ellipses indicate the same but for the COVID-19 period of 2020–2022.

related mortality from 70 heatwave days, calculated from the average of all identified heatwaves in the ten years that preceded COVID-19 (i.e., 2010–2019; see "Methods"). Substantially fewer heat-related deaths occurred during 70 heatwave days in 2010–2019, with the regional number ranging from 6 deaths per 100,000 people (95% confidence interval: 3–8) in North West England, to 14 deaths per 100,000 people (95% confidence interval: 12–15) in London. In other words, demand for regional health services was 1.6 (London) to 3.2-fold (North West England) when extreme heat coincided with COVID-19 in the study period, compared to the previous decade (Fig. 3e).

During the 70 cold snap days in the study period (30 January 2020 to 31 December 2022), regional compound (cold-related and COVID-19) mortality ranged from 80 per 100,000 people (95% confidence interval: 75–86) in Yorkshire and the Humber, to 127 deaths per 100,000 people (95% confidence interval: 123–132) in East of England (Fig. 3b), highlighting the higher absolute demand on the health system during cold snaps than heatwaves in the study period. These compound mortality numbers are substantially higher than the reference numbers of cold-related deaths from the same number of cold snap days in 2010–2019 (Fig. 3d), which ranged from 35 deaths per 100,000 people (95% confidence interval: 32–38) in London, to 48 deaths per 100,000 people (95% confidence interval: 43–55) in Wales. This means that depending on the region, demand for health services was 2 (South West England) to 3.4-fold (East of England) when extreme cold co-occurred with COVID-19, compared to extreme cold in the previous decade (Fig. 3f).

By adding temperature-related deaths (with COVID-19 deaths removed before calculation) and COVID-19 deaths together to estimate the compound mortality impact, we have assumed that they are independent of each other. Figure 4 shows the average number of temperature-related deaths per 100,000 population per day of individual heatwaves (panel a) and cold snaps (panel b) in all regions in 2016-2019 (non-COVID-19 years; grey markers) and 2020–2022 (COVID-19 years; coloured markers), versus the average temperatures of these events. While heat-related mortality generally increased with

the average heatwave temperature for all events, and cold-related mortality generally decreased when cold snaps were milder, the extreme temperature events that co-occurred with COVID-19 have different distributions (black solid ellipse contour) from the events that occurred without COVID-19 co-occurrence (grey dashed ellipse contour). For heatwaves (Fig. 4a), mean temperature and heat-related mortality shifted higher in the COVID-19 years, compared to events that were not affected by COVID-19. For cold snaps (Fig. 4b), the variances in temperature and cold-related mortality were larger in events in the COVID-19 years than in non-COVID-19 years. Two-sample Kolmogorov–Smirnov tests confirm that the COVID-19 event distributions are significantly different from the non-COVID-19 distributions at the 5% significance level. These results suggest that COVID-19 may have impacted temperature-related mortality during extreme weather events.

On the other hand, extreme heat may have exacerbated COVID-19 mortality in England and Wales too. This is evident on the hottest day ever recorded in the UK (19 July 2022), when 91 more daily COVID-19 deaths occurred, compared to the average on days between 10 July and 25 July 2022 (Fig. 1b). Separating the factors contributing to winter deaths is also challenging because low temperatures tend to be linked to influenza-like illnesses and respiratory diseases. Modelling COVID-19 mortality is not within the scope of this study, but our results highlight the complex interplay between extreme temperatures and the COVID-19 pandemic, as well as its implications on population health and health services capacity.

## Discussion

The 2023 UK National Risk Register states that extreme weather events and pandemics are two of the most serious risks facing the UK, alongside other risks such as terrorism, cyber-attacks and conflicts[32]. In this study, we have demonstrated the respective mortality impacts of extreme temperatures and COVID-19, as well as the substantial compound mortality impact arising from the co-occurrence of these two crises, on all regions in England and Wales. UK summers are becoming

hotter in a warming climate[33], and extreme heat events are increasing in frequency, intensity, and duration. While UK winters are generally becoming milder[33], extremely cold episodes driven by natural climate variability, such as the 2018 "Beast from the East", will still have a large adverse health impact. Viewing extreme heat and cold as serious health threats on par with the COVID-19 global pandemic and other major crises is an important first step to preventing further loss of lives to extreme weather.

With the latest long-term forecasts of COVID-19 for the UK predicting a steady 95% effective population immunity and a reduced death rate until at least May 2024[34], it is probable that cumulative temperature-related deaths would exceed COVID-19 deaths in more regions than South West England (Fig. 2) in the coming years. While extreme temperatures cannot be vaccinated against, interventions such as increasing tree coverage in cities[35], adding reflective or green roofs to buildings[36], installing external shutters to the housing stock[37], and air conditioning[38,39] have been shown to reduce the health risk of heat. In the colder months, heating homes has been shown to reduce cold-related mortality[39], and mechanical ventilation with heat recovery systems[40] can simultaneously improve insulation and energy efficiency, thereby reducing the health risk of cold, while maintaining ventilation. It is vital to implement these interventions in urban planning and housing regulations, in order to improve population resilience to heat and cold in the UK.

Recent analyses of NHS hospital beds show a persistent decline in the total number of beds, by 11% in England and 15% in Wales in the period 2010–2022[41,42]. Even before the COVID-19 pandemic, average hospital bed occupancy consistently surpassed 85%, a level widely considered to be the upper limit for hospitals to operate safely and effectively[43]. Many hospital trusts regularly exceeded an even higher capacity of 95% in winter[41]. These statistics highlight a significantly overwhelmed health system in England and Wales. The deadly consequences of an already overwhelmed NHS severely stretched to function through the compound crises of extreme weather and COVID-19, whose mortality impacts are additive to the first order, are clear in this paper. Decisive actions to improve NHS ability to substantially increase bed numbers and staff availability with little notice, since major crises tend to occur abruptly, are extremely important to build health resilience for future crises.

We have focused on England and Wales regions in this paper because the results are relevant to high-level decision-making in Government, as well as to raising general awareness of the health impacts of extreme heat and cold. A finer scale analysis is recommended for future work, as its results would be useful for, for example, counties to determine climate adaptation budgets and assess the costs and benefits of various climate and public health interventions.

Anticipating and preparing for the co-occurrence of extreme weather events and major disease outbreaks like COVID-19 would be lifesaving, not least because they share commonalities in who in the population are particularly vulnerable to them. For instance, extreme heat and cold adversely affect the health of older people (above the age of 65), young children (under the age of 5), people with pre-existing health conditions such as cardiovascular, respiratory and kidney conditions (kidney disease is a risk factor for heat-health), those who are socially isolated, those who are homeless, and those who live in deprived circumstances (a risk factor for cold-health), more than other population groups[31,44,45]. A number of these groups such as older people, people with various pre-existing conditions including severe asthma and chronic kidney disease, and those who are deprived are also at a higher risk of dying from COVID-19 in the UK[46,47]. Enhancing access to health and care services, support and guidance in regions with a highly vulnerable population (e.g., South West England and Wales both have over 20% of their population above the age of 65; Figure S2) and tailoring support for different vulnerable groups would simultaneously address their vulnerability to extreme weather and a

future coronavirus-type outbreak. Coronaviruses are still a priority pathogen with pandemic potential according to the WHO, along with Zika and diseases such as haemorrhagic fevers, influenza, and "disease X" (an unexpected new disease)[48].

Because of the common risk factors for temperature-related and COVID-19 mortality, disentangling their effects on health and attributing mortality to a single cause is challenging. Here, we have suggested that COVID-19 may have confounded with temperature-related mortality in England and Wales during heatwaves or cold snaps in the period 2020–2022 (Fig. 4), although the number of extreme temperature events sampled is limited by the short study period. This result is consistent with previous hypotheses that COVID-19 acted as a risk amplifier of heat-health impacts in the UK[22] and other countries including Portugal[49], potentially due to increased vulnerability, social restrictions, and people not seeking healthcare during the pandemic. On the contrary, another previous study found similar heatwave mortality in England in 2020 and the pre-pandemic period of 2016–2018, with no indication of more at-home heat-related deaths linked to COVID-19 social restrictions[50].

Future work is recommended to understand the confounding effect(s) of COVID-19 on temperature-related health outcomes. For example, instead of using all available observed temperature and mortality data (1981–2022 data, but with reported COVID-19 mortality removed) to construct the DLNMs here (see "Methods"), future work can derive temperature-mortality associations using only data from the COVID-19 time period. We have not done this because two years of data (in the COVID-19 period of 2020-2022) are not enough to derive constrained temperature-mortality associations with low uncertainty on the regional level. This limitation will be reduced when more data becomes available going forward. Future work can also investigate the feasibility of controlling for time-varying COVID-19 effects in the models.

On the other hand, we have shown above-average COVID-19 mortality in July 2022 when temperatures were unprecedentedly high (Fig. 1b). Detailed modelling of COVID-19 infections and mortality is out of the scope of this study, but future research on this topic is extremely important because the literature is inconclusive about the effect of temperature and humidity on the transmission of COVID-19, with some of these studies contradicting each other. For instance, previous studies found (i) low temperatures and/or low absolute humidity increased transmission[51], but only in the absence of policy interventions such as lockdowns[18]; (ii) modest non-linear associations between these variables and COVID-19 transmission[52]; and (iii) no association between temperature and COVID-19 but weak negative associations with absolute and relative humidity[53]. Recent research found that most of the studies about weather effects on COVID-19 spread that were published in 2020 and 2021, did not include a time lag between weather exposure and reported COVID-19 cases, did not properly account for seasonality or other confounding factors of disease transmission, or did not go through rigorous peer review[54]. These aspects need to be thoroughly considered in future research, for long-term planning and preparedness that are resilient to climate change.

## Methods
### Distributed lag nonlinear models (DLNMs)
For each region in England and Wales, we derived the non-linear association between daily average temperature and daily mortality in the period 1981–2022 (see Figure S3). We calculated daily average temperature by averaging the daily maximum and minimum temperatures in the HadUK-Grid Climate Observations by Administrative Regions over the UK dataset, which provided regional averages calculated from 1 km resolution gridded temperature data[55]. Weighting gridded HadUK-Grid temperatures by the 2021 Census population at the Lower layer Super Output Area level when computing the regional temperature averages did not change our main results (see Figures S4-S7).

We used daily all-cause death occurrences across all age and sex groups in the same time period. These data were provided by the Office for National Statistics (see Data availability). Since the 2020-2022 mortality data included COVID-19 deaths, which we wanted to separate from temperature-related deaths in this study, we removed deaths with COVID-19 on the death certificate from the all-cause mortality data. We then derived the non-linear association between the temperature and mortality time series using quasi-Poisson regression with DLNMs, adapting the methods from previous studies[56]. This time series regression model is shown in Eq. (1), where $E(Y_t)$ is the expected value of daily mortality at time $t$.

$$\log[E(Y_t)] = \alpha + f(x_t; l; \vartheta) + s(t; \boldsymbol{\beta}) \qquad (1)$$

Briefly, we modelled the temperature-mortality association, $f(x_t; l; \vartheta)$ in Eq. 1, for each region using natural cubic splines with three internal knots at the 10th, 75th, and 90th percentiles of the 1981–2022 temperature distribution of that region. Since we were interested in both heat- and cold-related mortality, year-round temperatures ($x_t$) were considered. We used the same splines function[56] and internal knots[24,56] as previous epidemiological studies that had tested these modelling choices on year-round UK data. It is known that the mortality effect of exposure to heat and cold can take up to a few days (heat) and a few weeks (cold) to be observed[56], so we included a time lag association in each DLNM ($l$ denotes the lag time). We modelled this association with natural cubic splines with three internal knots that are equally spaced in the log scale, following a standard set up in the literature[24]. We considered a maximum lag period of 21 days, which is sufficient to capture cold health effects[24].

In the main quasi-Poisson regression model, we also included an indicator of the day of the week and a natural cubic spline of time with 8 degrees of freedom per year ($s(t; \boldsymbol{\beta})$ in Eq. 1). This again followed a tested set-up in the literature[27] and controlled for residual variation and seasonality in the data. Through the methods described above, we found the overall temperature-mortality association across the 21-day lag period for each region in England and Wales. This overall association was centred on a 'minimum mortality temperature' (MMT) corresponding to the region. We constrained the MMT to the 2nd to 98th percentiles of the 1981–2022 daily average temperature distribution of the region. By definition, mortality risk at the MMT is the lowest among all observed temperatures of the same region. Increased mortality risk is expected above (considered as heat-related mortality) and below (cold-related mortality) the MMT.

Based on the observed daily average temperature between 30 January 2020 and 31 December 2022 from HadUK-Grid, the derived mortality risk at each temperature, and the average number of all-cause deaths (excluding COVID-19 deaths) for each calendar day in a year based on the 1981-2022 period, best estimates of temperature-related mortality for each day in the study period were calculated for each region. We categorised the resulting temperature-related mortality as heat-related if that day's regional mean temperature was above the corresponding MMT, and vice versa for cold-related mortality. Their 95% empirical confidence intervals were estimated through Monte Carlo simulations of the spline model coefficients 100 times, assuming that the coefficients had a multivariate normal distribution.

### Heatwaves and cold snaps before 2016
Throughout this study, we focused on heatwaves and cold snaps when extreme temperatures have the largest mortality impacts. The dates of these events were extracted from the UKHSA Heatwave Mortality Reports and historical Level 3 Cold Health Alerts, as mentioned in the main text. However, these reports and the alerts record available to us only started in 2016, so we found the dates of heatwaves and cold snaps based on the same criteria for the period 2010–2015 for Fig. 3. For heatwaves, we found days on which there was a UKHSA Level 3

Heat Health Alert, or days with a mean central England temperature greater than 20 °C, plus one day before and after the time period identified through the first two criteria. The 2010–2015 heatwaves defined this way were published in the literature[22]. For cold snaps, we found periods during which any region in England and Wales had an average temperature below 2 °C for two consecutive days or more. This follows the temperature criterion for a Level 3 Cold Health Alert in 2016[57].

### Deaths per 100,000 population
We present the number of deaths (temperature-related or COVID-19) per 100,000 population in the latter part of this manuscript to enable comparison of the results across the studied regions. For the study period between 30 January 2020 and 31 December 2022, these were calculated based on the mid-2021 UK population estimates provided by the Office for National Statistics (see Data availability). The regional population data were as follows – North East England: 2,646,772; North West England: 7,422,295; Yorkshire and the Humber: 5,481,431; East Midlands: 4,880,094; West Midlands: 5,954,240; East of England: 6,348,096; London: 8,796,628; South East England: 9,294,023; South West England: 5,712,840; and Wales: 3,105,410. For the period 2010-2019, the mid-2015 regional population estimates – North East England: 2,624,579; North West England: 7,175,178; Yorkshire and the Humber: 5,390,211; East Midlands: 4,677,425; West Midlands: 5,755,032; East of England: 6,075,970; London: 8,666,930; South East England: 8,949,392; South West England: 5,471,610; and Wales: 3,099,086– were used instead.

### Reporting summary
Further information on research design is available in the Nature Portfolio Reporting Summary linked to this article.

## Data availability
Met Office HadUK-Grid climate observations by administrative regions over the UK are available on the CEDA archive (https://catalogue. ceda.ac.uk/uuid/b39898e76ab7434a9a20a6dc4ab721f0). Daily all-cause death occurrences in regions of England and Wales are available from the Office for National Statistics (https://www.ons.gov.uk/ peoplepopulationandcommunity/birthsdeathsandmarriages/deaths/ adhocs/14173dailydeathsoccurrencesenglandandwales1981and2020 and https://www.ons.gov.uk/peoplepopulationandcommunity/birthsdeaths andmarriages/deaths/adhocs/1724dailydeathoccurrencesenglandand wales2021and2022). Mid-2021 and mid-2015 population estimates for the UK are available from the Office for National Statistics (https://www. ons.gov.uk/peoplepopulationandcommunity/populationandmigration/ populationestimates/datasets/populationestimatesforukenglandand walesscotlandandnorthernireland). Daily deaths with COVID-19 on the death certificate by UK regions and nations can be obtained from the Government Coronavirus (COVID-19) in the UK dashboard (https:// coronavirus.data.gov.uk/details/deaths). Cumulative COVID-19 vaccine uptake by dosage and regions was available on the dashboard (https:// coronavirus.data.gov.uk/details/vaccinations), but it has since been discontinued. A copy of this discontinued data is available at this paper's GitHub repository (https://github.com/BrisClimate/Compound-temps-covid-mortality-paper). NHS England hospital bed availability and occupancy are available from NHS England Statistics (https://www.england. nhs.uk/statistics/statistical-work-areas/bed-availability-and-occupancy/). NHS Wales hospital bed availability and occupancy are available from StatsWales (https://statswales.gov.wales/Catalogue/Health-and-Social-Care/NHS-Hospital-Activity/NHS-Beds).

## Code availability
All code used to generate the results and figures in this study is publicly available on GitHub at https://github.com/BrisClimate/Compound-temps-covid-mortality-paper.

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

## Acknowledgements

This research is a direct result of a discussion with Sir Patrick Vallance, then Chief Scientific Adviser to the UK Government, at COP26 in 2021. We thank Sir Patrick for a follow-up meeting at the Natural History Museum in 2023. We thank Prof Oliver Johnson and Dr Juliette Unwin of the University of Bristol for discussing COVID-19 statistics with us. We thank Prof Philip Taylor of the University of Bristol for supporting this research. We thank Dr Emily Ball and Dr Chin Yang Shapland of the University of Bristol for discussing the technical aspects of the revised manuscript. Y.T.E.L. was funded by the University of Bristol Climate Change and Health Fellowship. D.M.M was supported by the NERC Grant ArctiCONNECT (Grant ID: NE/V005855/1). A.G. was supported by the Medical Research Council-UK (Grant ID: MR/V034162/1).

## Author contributions

D.M.M. conceived the research idea. Y.T.E.L. conducted the analysis, made the figures, and wrote the initial manuscript. D.M.M. and A.G. provided comments on the figures and manuscript.

## Competing interests

The authors declare no competing interests.
