## [Peer Review File · Nature Communications]

Compound mortality impacts from extreme temperatures and the COVID-19 pandemicREVIEWER COMMENTS

Reviewer #1 (Remarks to the Author):

This is an original work using high-quality data and state-of-the-art methodology. My comments for the authors are provided below.

Results section:

1. L122-123. I would delete "in distributed lag non-linear models (DLNMs) for health modelling".
2. L226-228 and L239-240. The authors use a mortality ratio as an indicator of health services demand. I am not sure whether this is a valid and recognized indicator for monitoring health services demand. I would rewrite the sentences under consideration.
3. Figure 3. I would add a third row of maps showing the ratio a/c and b/d, and reference these new maps in the text (L226-228 [heat] and L239-240 [cold]).
4. L250-253. I think the sentence should read: "the 250 extreme temperature events that co-occurred with COVID-19 have different distributions (black solid lines) from the events that occurred without COVID-19 co-occurrence (grey dashed lines)". Moreover, I would replace the term "lines" by "ellipse contour".

Discussion section:

5. L296-303. It has been reported in the literature that air conditioning and heating are effective adaptive measures to high and low temperatures:

- Sera, F. et al. Air conditioning and heat-related mortality: A multi-country longitudinal study. *Epidemiology* 31, 779–787 (2020).

- Achebak, H. et al. Drivers of the time-varying heat-cold-mortality association in Spain: A longitudinal observational study. *Environ. Int.* 182, 108284 (2023).

Methods section:

6. I think the methods should be reported using the past tense.
7. Mortality data were not available at a finer geographical scale? (eg, Counties)
8. What is the spatial resolution of the HadUK-Grid?
9. Temperature values should be weighted by population density as regions in England and Wales are large.
10. Some modelling choices are not sufficiently justified (eg, number degrees of freedom [DF] for seasonality) and the sensitivity analysis is missing (eg, DF for seasonality, internal knots for the exposure-response function).
11. L386-398. The authors could improve the description the time-series regression model, and especially, the DLNM.
12. I would include the formula of the time-series regression model.
13. I would report the region-specific temperature-mortality association (ie, risk curves) in the Supplement.
14. L410-411. How did you calculate heat- and cold-related mortality?

Reviewer #2 (Remarks to the Author):

In this study, Dr. Eunice Lo et al. used daily temperature and mortality data from 1981 to 2022 in England and Wales to compare the excess mortality attributable to COVID-19 and extreme temperatures. They found that mortality attributable to extreme temperatures exceeded that attributable to COVID-19 during heatwaves in the majority of UK regions, while this was not the case during cold snaps. It is worth noting that this finding comes as no surprise, as viral infections typically lead to increased mortality only during the winter months, while reduced circulation in spring causes minimal harm. Below are some comments aimed at improving the manuscript:

1. Line 61: Suggest replacing "social restriction" with "physical distancing."

2. Lines 165-167: The statement "These results demonstrate that even during a global pandemic that dominated everyone's lives, heat and cold were a comparable but arguably much more under-reported killer than COVID-19" appears unsupported by the data. The data indicates that, over the entire period, COVID-19 mortality generally exceeded mortality attributable to extreme temperatures in UK regions, with the exception of South West England. Therefore, the suggestion of under-reporting seems unfounded.

3. Lines 319-325: The authors rightly acknowledged that extreme temperatures mainly affect the health of older people, and this holds true for COVID-19 as well, as it tends to be more severe in the older population. However, the study did not consider the varying age structures across UK regions. For instance, in 2021, the median age in the South West was 44 years, while in London, it was 35 years (source: <https://www.ons.gov.uk/peoplepopulationandcommunity/populationandmigration/populationestimates/bulletins/populationandhouseholdestimatesenglandandwales/census2021unroundeddata>). This may introduce bias into the comparison of excess mortality across UK regions. Therefore, it is recommended that results be presented as age-adjusted rates.

Reviewer #2 (Remarks on code availability):

I have replicated the code provided by the authors and successfully obtained the expected results. All required documentation and metadata were provided.

Reviewer #3 (Remarks to the Author):

Overall, this is a very good study – an important topic and well suited design. I have some comments but I view all of them as minor and they require authors only clarify or add some context. I recommend authors revise and resubmit.

1. In the comparison between several extreme weather events and their effect, authors should emphasize that in low-capacity settings such as Pakistan and Africa – where vital registration is very lacking, the figures aren't comparable to UK estimated excess deaths where vital registration is complete and causes of death are properly attributed. See estimates of death registration completeness by UNSD <https://unstats.un.org/unsd/demographic-social/crvs/> and proper attribution of deaths by <https://doi.org/10.1186/s12911-021-01501-1>.

2. In the paragraph on covid-19 deaths, authors cite 7M deaths, but this is only officially reported deaths, with excess deaths from covid being much higher, especially in developing countries. Seeing that this is a paper on excess mortality, this is highly relevant. Authors can cite the WHO's excess deaths estimate study on this: <https://www.nature.com/articles/s41586-022-05522-2>.

3. When discussing results for all of England & Wales (i.e. figure 1), it's unclear to me if the results are the sum of the regional-level models or a model fitted on the total itself.

4. Authors compare climate deaths per capita by region for 2010-2019 vs. study period (2020-2022), but in the Methods section cite regional-population counts only for 2021. What population counts were used for 2010-2019? This should be added and more clearly described.

5. Authors rightfully point out that cold-related deaths are harder to ascertain than heat-related deaths, and that they come with a longer lag. Another important reason is that low-temperatures carry with it many ILI and respiratory diseases, making it hard to distinguish between these factors.

Reviewer #1 (Remarks to the Author):

This is an original work using high-quality data and state-of-the-art methodology. My comments for the authors are provided below.

Results section:

1. L122-123. I would delete “in distributed lag non-linear models (DLNMs) for health modelling”.

We have deleted this part of the sentence (line 145), thank you.

2. L226-228 and L239-240. The authors use a mortality ratio as an indicator of health services demand. I am not sure whether this is a valid and recognized indicator for monitoring health services demand. I would rewrite the sentences under consideration.

Thank you for raising this point. In these lines (now 266-268, 288-289), we describe results in Figure 3, which shows the total number of deaths from non-optimal temperatures and COVID-19. We think these death tolls are indicative of the burden on health services, based on evidence in the literature that deaths in hospitals, hospices and care homes tend to be above average during heatwaves (Thompson et al., 2022; ONS, 2022), and that hospitals and care homes are the most common places of excess winter deaths in England and Wales (ONS, 2021). The ratios that the reviewer mentioned are comparisons of the death tolls between the study period and the decade before that. We expect that higher death tolls in the study period mean higher health services demand.

Thompson et al., 2022 is reference 56 in the revised manuscript.

ONS, 2022:

<https://www.ons.gov.uk/peoplepopulationandcommunity/birthsdeathsandmarriages/deaths/articles/excessmortalityduringheatperiods/englandandwales1juneto31august2022>

ONS, 2021:

<https://www.ons.gov.uk/peoplepopulationandcommunity/birthsdeathsandmarriages/deaths/bulletins/excesswintermortalityinenglandandwales/2020to2021provisionaland2019to2020final>

3. Figure 3. I would add a third row of maps showing the ratio a/c and b/d, and reference these new maps in the text (L226-228 [heat] and L239-240 [cold]).

This is an excellent idea. We have edited Figure 3 and added cross-referencing in the text (lines 268 and 290).

4. L250-253. I think the sentence should read: “the 250 extreme temperature events that co-occurred with COVID-19 have different distributions (black solid lines) from the events that occurred without COVID-19 co-occurrence (grey dashed lines)”. Moreover, I would replace the term “lines” by “ellipse contour”.

Thank you for spotting this, the sentence has been corrected (line 301-302).

Discussion section:

5. L296-303. It has been reported in the literature that air conditioning and heating are effective adaptive measures to high and low temperatures:

- Sera, F. et al. Air conditioning and heat-related mortality: A multi-country longitudinal study. *Epidemiology* 31, 779–787 (2020).

- Achebak, H. et al. Drivers of the time-varying heat-cold-mortality association in Spain: A longitudinal observational study. *Environ. Int.* 182, 108284 (2023).

Absolutely, we have cited these papers in line 355-356 now.

Methods section:

6. I think the methods should be reported using the past tense.

We have changed it to past tense, thank you.

7. Mortality data were not available at a finer geographical scale? (eg, Counties)

We appreciate this question. Since the overarching aim of this paper is to communicate the importance of extreme weather on public health to policymakers and the public, and that heatwaves and cold snaps often occur at spatial scales that are as least as large as the studied regions, we consider the regional scale to be appropriate for this paper. A finer scale analysis would be beneficial for, for example, climate adaptation budget and cost-benefit analysis in counties, so we suggest this as future work in the revised manuscript. This suggested work would require fine-scale mortality data that are not publicly available.

Line 384-489 now reads “We have focused on England and Wales regions in this paper because the results are relevant to high-level decision making in Government, as well as to raising general awareness of the health impacts of extreme heat and cold. A finer scale analysis is recommended for future work, as its results would be useful for, for example, counties to determine climate adaptation budget and assess the costs and benefits of various climate and public health interventions”.

8. What is the spatial resolution of the HadUK-Grid?

The HadUK-Grid Climate Observations by Administrative Regions over the UK data were area averages calculated from 1 km gridded data. This has been clarified in line 475-476.

9. Temperature values should be weighted by population density as regions in England and Wales are large.

Thank you for your comment. We have repeated the analysis by weighting 25 km gridded HadUK-Grid temperature data, by Lower layer Super Output Area (LSOA) level population data from the 2021 Census. There were 35,672 LSOAs in England and Wales, so this data represented fine-scale population. For each 25 km grid cell, we found all LSOAs whose centres were within the grid boundary, and we added the population of these LSOAs together to find the population of that grid cell. We chose the 25 km grid in order to avoid double counting population in LSOAs that spanned more than one grid cell (LSOAs are irregular shapes). We then found all 25 km grid cells within each studied region, and calculated the population weighted average temperature from them.

Weighting the temperature data by population has not changed our main results at all. For this reason, we have kept our original results in the main manuscript, but we have added the new results to Supplementary Information (Figures S4-S7) as a sensitivity test. These are now mentioned in line 476-478 in the main manuscript.

10. Some modelling choices are not sufficiently justified (eg, number degrees of freedom [DF] for seasonality) and the sensitivity analysis is missing (eg, DF for seasonality, internal knots for the exposure-response function).

Thanks for raising this point. These modelling choices had been tested and justified in UK analyses in previous studies in the climate epidemiology literature. We have added clarification and the relevant citations to lines 498-500 and 510.

11. L386-398. The authors could improve the description the time-series regression model, and especially, the DLNM.

We hope the added justification of modelling choices and equation (see below) have improved the description of the time series regression model.

12. I would include the formula of the time-series regression model.

Equation 1 has been added to Methods (line 492-494), with its terms referenced in the text that follows.

13. I would report the region-specific temperature-mortality association (ie, risk curves) in the Supplement.

These are now in Figure S3 and mentioned in line 472 in Methods, thank you for your suggestion.

14. L410-411. How did you calculate heat- and cold-related mortality?

This has now been clarified in line 530-536, which reads “Based on the observed daily average temperature between 30 January 2020 and 31 December 2022 from HadUK-Grid, the derived mortality risk at each temperature, and the average number of all-cause deaths (excluding COVID-19 deaths) for each calendar day in a year based on the 1981-2022 period, best estimates of temperature-related mortality for each day in the study period were calculated for each region. We categorised the resulting temperature-related mortality as heat-

related if that day's regional mean temperature was above the corresponding MMT, and vice versa for cold-related mortality.”

Reviewer #2 (Remarks to the Author):

In this study, Dr. Eunice Lo et al. used daily temperature and mortality data from 1981 to 2022 in England and Wales to compare the excess mortality attributable to COVID-19 and extreme temperatures. They found that mortality attributable to extreme temperatures exceeded that attributable to COVID-19 during heatwaves in the majority of UK regions, while this was not the case during cold snaps. It is worth noting that this finding comes as no surprise, as viral infections typically lead to increased mortality only during the winter months, while reduced circulation in spring causes minimal harm. Below are some comments aimed at improving the manuscript:

1. Line 61: Suggest replacing "social restriction" with "physical distancing."

This has been changed (line 63), thank you.

2. Lines 165-167: The statement "These results demonstrate that even during a global pandemic that dominated everyone's lives, heat and cold were a comparable but arguably much more under-reported killer than COVID-19" appears unsupported by the data. The data indicates that, over the entire period, COVID-19 mortality generally exceeded mortality attributable to extreme temperatures in UK regions, with the exception of South West England. Therefore, the suggestion of under-reporting seems unfounded.

We appreciate this comment. We have changed this sentence to "These results demonstrate the importance of increasing public health messaging about heat and cold, which tends to be far less prevalent than the messaging about COVID-19" (line 191-192).

3. Lines 319-325: The authors rightly acknowledged that extreme temperatures mainly affect the health of older people, and this holds true for COVID-19 as well, as it tends to be more severe in the older population. However, the study did not consider the varying age structures across UK regions. For instance, in 2021, the median age in the South West was 44 years, while in London, it was 35 years (source: <https://www.ons.gov.uk/peoplepopulationandcommunity/populationandmigration/populationestimates/bulletins/populationandhouseholdestimatesenglandandwales/census2021unroundeddata>). This may introduce bias into the comparison of excess mortality across UK regions. Therefore, it is recommended that results be presented as age-adjusted rates.

This is a good point, thank you. We discussed age and other vulnerability factors to extreme temperatures and COVID-19 in the original manuscript. We believe that a comparison across regions using non-age standardised rates offers a more realistic picture of the mortality impacts. However, we recognise the reviewer's point and have now added a map showing regional percentages of population aged above 65 to Supplementary Information (Figure S2). South West England and Wales have over 20% of their population aged above 65, while London has the lowest percentage. We think this information is important for policy interventions. We have added "Enhancing access to health and care

services, support and guidance in regions with a high vulnerable population (e.g., South West England and Wales both have over 20% of their population being above the age of 65; Figure S2) and tailoring support for different vulnerable groups would simultaneously address their vulnerability to extreme weather and a future coronavirus-type outbreak” (line 409-413) to Discussion.

Reviewer #2 (Remarks on code availability):

I have replicated the code provided by the authors and successfully obtained the expected results. All required documentation and metadata were provided.

We are pleased to hear this!

Reviewer #3 (Remarks to the Author):

Overall, this is a very good study – an important topic and well suited design. I have some comments but I view all of them as minor and they require authors only clarify or add some context. I recommend authors revise and resubmit.

1. In the comparison between several extreme weather events and their effect, authors should emphasize that in low-capacity settings such as Pakistan and Africa – where vital registration is very lacking, the figures aren’t comparable to UK estimated excess deaths where vital registration is complete and causes of death are properly attributed. See estimates of death registration completeness by UNSD <https://unstats.un.org/unsd/demographic-social/crvs/> and proper attribution of deaths by <https://doi.org/10.1186/s12911-021-01501-1>.

Thank you for pointing this out. The sentence “These places have different levels of death registration completeness and cause of death accuracy, so these numbers cannot be compared like-for-like” and the suggested citations have been added to line 47-49.

2. In the paragraph on covid-19 deaths, authors cite 7M deaths, but this is only officially reported deaths, with excess deaths from covid being much higher, especially in developing countries. Seeing that this is a paper on excess mortality, this is highly relevant. Authors can cite the WHO’s excess deaths estimate study on this: <https://www.nature.com/articles/s41586-022-05522-2>.

This is indeed very relevant to this paper. The global COVID-19 deaths number has been replaced by the one estimated in the recommended paper. The sentence now reads “In the two-year period of 2020-2021, almost 15 million excess deaths were estimated to be associated with COVID-19 directly or indirectly globally” (line 57-59).

3. When discussing results for all of England & Wales (i.e. figure 1), it’s unclear to me if the results are the sum of the regional-level models or a model fitted on the total itself.

It is the sum of the regional models. We have added “All temperature-related deaths here represent the sum of regional deaths estimated from individual temperature-mortality associations for ten regions in England and Wales (see

Methods)” to the caption of Figure 1 for clarity.

4. Authors compare climate deaths per capita by region for 2010-2019 vs. study period (2020-2022), but in the Methods section cite regional-population counts only for 2021. What population counts were used for 2010-2019? This should be added and more clearly described.

Thank you for raising this important point. We had not accounted for population change between the two periods in the original manuscript. We have now corrected this using the mid-2015 regional population estimates (from the Office for National Statistics) for the period 2010-2019. We have (i) updated Figures 3 and 4 (and the new Figures S6 and S7 in Supplementary Information), (ii) updated the main text that interprets these figures (lines 264-267, 277-290), and (iii) added the mid-2015 population estimates to Methods (line 570-574). The main results have not changed by incorporating the population change.

5. Authors rightfully point out that cold-related deaths are harder to ascertain than heat-related deaths, and that they come with a longer lag. Another important reason is that low-temperatures carry with it many ILI and respiratory diseases, making it hard to distinguish between these factors.

This is true. We have added “Separating the factors contributing to winter deaths is also challenging because low temperatures tend to be linked to influenza-like illnesses and respiratory diseases” to line 329-331.

REVIEWERS' COMMENTS

Reviewer #1 (Remarks to the Author):

The authors have successfully addressed all my comments. Thank you.

Reviewer #3 (Remarks to the Author):

Authors have more than sufficiently responded to the issues and questions I raised in this revision, and I'm glad that I was able to help catch some (albeit minor) issues in the previous version. I recommend acceptance.